

**Heterogeneous photochemistry of imidazole-2-**
**carboxaldehyde: HO₂ radical formation and aerosol growth**
**Laura González Palacios[1,2], Pablo Corral Arroyo[3,4], Kifle Z. Aregahegn,[5,6] Sarah S.**
**Steimer[3,4,7], Thorsten Bartels-Rausch[3], Barbara Nozière,[5] Markus Ammann[3,4],**
**Christian George[5] and Rainer Volkamer[1,2]**
[1]{University of Colorado, Department of Chemistry and Biochemistry, 215 UCB, Boulder,
CO, 80309, USA}
[2]{University of Colorado, Cooperative Institute for Research in Environmental Sciences
(CIRES), 216 UCB, Boulder, CO, 80309, USA}
[3]{Paul Scherrer Institute, Laboratory of Radio- and Environmental Chemistry, 5232 Villigen
PSI, Switzerland}
[4]{Swiss Federal Institute of Technology Zurich, Institute for Atmospheric and Climate
Science, 8092 Zürich, Switzerland}
[5]{Université Lyon 1; Centre National de la Recherche Scientifique (CNRS), UMR5256,
IRCELYON, Institut de recherches sur la catalyse et l'environnement de Lyon, F-69626
Villeurbanne, France}
[6]{now at: Chemistry Department, University of California, Irvine, California, 92697-202 }
[7]{now at: University of Cambridge, Department of chemistry, Cambridge CB2 1EW, UK}
Correspondence to: R. Volkamer (rainer.volkamer@colorado.edu).
**Abstract**
The multiphase chemistry of glyoxal is a source of secondary organic aerosol (SOA), including its
light-absorbing product imidazole-2-carboxaldehyde (IC). IC is a photosensitizer that can
contribute to additional aerosol ageing and growth when its excited triplet state oxidizes
hydrocarbons (reactive uptake) via H-transfer chemistry. We have conducted a series of
photochemical coated-wall flow tube (CWFT) experiments using films of IC and citric acid (CA),
an organic proxy and H-donor in the condensed-phase. The formation rate of gas-phase $HO_2$





radicals ($P_{HO2}$) was measured indirectly by converting gas-phase NO into $NO_2$. We report on
experiments that relied on measurements of $NO_2$ formation, NO loss; and HONO formation. $P_{HO2}$
was found to be a linear function of (1) the [IC]×[CA] concentration product, and (2) the photon
actinic flux. Additionally, (3) a more complex function of relative humidity (25% < RH < 63%),
and of (4) the $O_2/N_2$ ratio (15% < $O_2/N_2$ < 56%) was observed, most likely indicating competing
effects of dilution, $HO_2$ mobility and losses in the film. The maximum $P_{HO2}$ was observed at 25-
55% RH and at ambient $O_2/N_2$. The $HO_2$ radicals form in the condensed-phase when excited IC
triplet states are reduced by H-transfer from a donor, CA in our system, and subsequently react
with $O_2$ to re-generate IC, leading to a catalytic cycle. OH does not appear to be formed as a
primary product but is produced from the reaction of NO with $HO_2$ in the gas phase. Further, seed
aerosols containing IC and ammonium sulfate were exposed to gas-phase limonene and $NO_x$ in
aerosol flow tube experiments, confirming significant $P_{HO2}$ from aerosol surfaces. Atmospheric
implications consist in a potentially relevant contribution of triplet state photochemistry for gas-
phase $HO_2$ production, aerosol growth and ageing.

## 1. Introduction

The sources and sinks of radicals play an important role in the oxidative capacity of the
atmosphere. Radicals and other oxidants initiate the chemical degradation of various trace gases,
which is key in the troposphere (Jacob, 1999). The hydroxyl (OH) and peroxyl ($HO_2$) radicals
belong to the $HO_x$ chemical family and are primarily generated by ultraviolet radiation
photochemical reactions (Calvert and Pitts, 1966), like the reaction of $O(^1D)$ (from $O_3$) with $H_2O$,
or photolysis of HONO, HCHO, $H_2O_2$, or acetone. Some secondary gas-phase sources are the
ozonolysis of alkenes or $O(^1D)$ + $CH_4$ (Monks, 2005). The oxidation of VOCs by OH and other
oxidants in the presence of NO leads to perturbations in the $HO_x$, $NO_x$, and $RO_x$ radical cycles that
affect $O_3$ and aerosol formation (Monks, 2005; Sheehy et al., 2010). The kinetics and
photochemical parameters of these reactions are relatively well-known in the gas-phase (Atkinson
et al., 2004; Sander et al., 2011). However, this does not apply to the sources and sinks for $HO_x$
in atmospheric droplets and on aerosol surfaces (Ervens et al., 2011). Uptake of OH from the gas-
phase, and $H_2O_2$ photolysis in the condensed phase are the primary known sources for $HO_x$ in the
condensed-phase. $HO_2$ is highly soluble and the concentrations of OH, the most effective oxidant





in the condensed phase, depend on $HO_2$. Another source of $HO_x$ radicals is from the chemical
reactions of reduced metal ions and $H_2O_2$, known as Fenton reactions (Fenton, 1894; Deguillaume
et al., 2005). Direct photolysis of $H_2O_2$, nitrite, nitrate (Zellner et al., 1990), hydroperoxides (Zhao
et al., 2013), and light absorbing secondary organic aerosol (SOA) (Badali et al., 2015) are also
sources of $HO_x$ in the condensed-phase. Other studies have shown that the photochemistry of iron
(III) oxalate and carboxylate complexes, present in aqueous environments (e.g. wastewater,
clouds, fogs, particles), can initiate a radical chain reaction serving as an aqueous source of $HO_2$
and $Fe^{2+}$. $Fe^{2+}$ can then regenerate OH starting a new cycle of Fenton reactions (Weller et al.,
2013a, 2013b). The temperature dependent rate constants of OH in the aqueous phase have been
studied for a limited subset of organics (Ervens et al., 2003). However, there is still a wide gap
with respect to understanding the sources, sinks, kinetics and photochemical reaction pathways of
$HO_x$ radicals in the condensed phase (George et al., 2015).
Our study investigates photosensitizers as an additional $HO_x$ source that may be relevant to further
modify $RO_x$ and $NO_x$ reaction cycles in both the condensed- and gas-phases. It is motivated by
the formation of superoxide in terrestrial aqueous photochemistry (Draper and Crosby, 1983;
Faust, 1999; Schwarzenbach et al., 2002), by more recent observations that irradiated surfaces
containing titanium dioxide generate $HO_x$ radicals in the gas-phase (Yi et al., 2012) and by the
generation of OH from metal oxides acting as photocatalysts in mineral dust (Dupart et al., 2012).
Past studies have demonstrated the reactivity of glyoxal towards ammonium ions and amines as a
source for light-absorbing brown carbon (Nozière et al., 2009; Galloway et al., 2009; Shapiro et
al., 2009; Kampf et al., 2012). One of these products is imidazole-2-carboxaldehyde (IC)
(Galloway et al., 2009), which absorbs light at UV wavelengths ($\lambda < 330$ nm) (Maxut et al., 2015).
Other imidazole-type compounds and light-absorbing products are formed in minor amounts but
can nonetheless impact optical and radiative properties of SOAs (Sareen et al., 2010; Trainic et
al., 2011). Photochemical reactions by these species are not typically accounted for in models yet,
but have a possible role for SOA formation and aerosol aging mechanisms (Sumner et al., 2014).
Photosensitizers are light absorbing compounds that absorb and convert the energy of photons into
chemical energy that can facilitate reactions, e.g., at surfaces or within aerosols (George et al.,
2015). For example, aerosol seeds containing humic acid or 4-(benzoyl)benzoic acid (4-BBA),
two other known photosensitizers, can induce the reactive uptake of VOCs when exposed to light,
leading to secondary organic aerosol (SOA) formation (Monge et al., 2012). Aregahegn et al.





(2013) and Rossignol et al. (2014) suggested a mechanism for autophotocatalyic aerosol growth,
where radicals are produced from the reaction of an H-donor hydrocarbon species, in this case
limonene, and the triplet state of IC.  Field measurements on fog water samples confirmed that
triplet excited states of organic compounds upon irradiation can oxidize model samples such as
syringol (a biomass burning phenol) and methyl jasmonate (a green leaf volatile), accounting for
30 – 90% of their loss (Kaur et al., 2014).
The existence of such photocatalytic cycles could be of atmospheric significance.  Canonica et al.
(1995) suggested indeed that the initial carbonyl, triggering the photochemical properties, is
regenerated via a reaction with oxygen producing $HO_2$.  To our knowledge, the production of such
radical side products was not investigated under atmospheric conditions previously.  We therefore
report here on the $HO_2$ radical production from IC in the condensed-phase.

**2. Experimental Section**
A series of flow tube experiments were conducted to investigate the formation of gas-phase $HO_2$
radicals from IC photochemistry using two different CWFT reactors (Sect. 2.1).  Section 2.2
describes aerosol flow tube experiments that in absence of other known radical sources confirm
that $HO_2$ production from aerosols can start photochemistry. All experiments were performed at
atmospheric pressure.
**2.1. Coated-wall flow tube experiments**
The CWFT experiments were designed to investigate the gas-phase production of $HO_2$ radicals
from a film containing IC and citric acid (CA) matrix as a function of UV light intensity, IC
concentration in the film, relative humidity (RH), and $O_2$ mixing ratio. Two similar experimental
setups were used as shown in Fig. 1. Some of the differences, not major, consist in the flow reactor
volume, surface area, flow rates, IC mass loading, NO mixing ratio, temperature inside the reactor
and the connected instrumentation.
**Setup 1.** Experiments were conducted in a photochemical flow-system equipped with a Duran
glass CWFT (0.40 cm inner radius, 45.2 and 40.0 cm length, inner surface = 113.6 and 100.4 $cm^2$,
S/V = 5.00 $cm^{-1}$), which was housed in a double jacketed cell coupled to a re-circulating water
bath to control the temperature at 298 K; The setup is shown in Fig. 1A. A thin film of IC+CA was





deposited inside the tubular glass flow tube. The experimental procedure for the preparation of
the films is described in Sect. 2.1.2. The system consisted of seven ultraviolet lamps (UV-A range,
Philips Cleo Effect 22 W: 300-420 nm, 41 cm, 2.6 cm o.d.) surrounding the flow tube in a circular
arrangement of 10 cm in diameter.
**Setup 2.** The second CWFT (CWFT 0.60 cm inner radius, 50 cm length, inner surface 188.5 cm$^2$,
S/V = 3.33 cm$^{-1}$) reactor had a glass jacket to allow water to circulate and maintain temperature
control inside the tube at 292 K. The coated-wall tubes were snuggly fit into the CWFT as inserts.
The CWFT was surrounded by the same seven fluorescent lamps as in Setup 1. The light passed
through different circulating water cooling jackets for both setups, thus providing a different light
path for each setup.
**Setup 1 and 2.** The actinic flux in the flow tube reactor, $F_{FT}(\lambda)$, was measured by actinometry of
NO$_2$ (see Supplement for description of $J_{NO2}$ measurements), independently for both setups. The
flows of N$_2$, O$_2$, air and NO were set by mass flow controllers. The RH was set by a humidifier
placed after the admission of N$_2$ and O$_2$ gases but before the admission of NO or NO$_2$ (see Fig. 1),
in which the carrier gas bubbles through liquid water at a given temperature. The humidifier could
also be by-passed to set a RH of near zero. A typical measurement sequence is described in Sect.

134   2.1.2.

The $J_{NO2}$ was measured for both Setup 1 and 2 using NO$_2$ actinometry. The $J_{NO2}$ with seven lamps
was found to be $2 \times 10^{-2}$ s$^{-1}$ for Setup 1 and $1 \times 10^{-2}$ s$^{-1}$ for Setup 2 (see Fig. S2 for Setup 1, and
Supplemental Information text for both Setups). These values were compared to direct irradiance
measurements in the flow tube and thus normalized (see Sect. 3.1.1).

## 2.1.1. Flow tube instrumentation

The following gas-phase products exiting the flow tube were measured by three different
instruments: NO$_2$ by the University of Colorado Light Emitting Diode Cavity-Enhanced
Differential Optical Absorption Spectroscopy (LED-CE-DOAS) instrument (Thalman and
Volkamer, 2010), HONO by a LOng Path Absorption Photometer (LOPAP, QuMA Gmbh,
Heland, J., 2001; Kleffmann et al., 2002), and NO by a chemiluminescence analyzer (Ecophysics
CLD 77 AM, also used for NO$_2$ in Setup 2). HO$_2$ radicals were indirectly measured by detecting



NO$_2$ with the LED-CE-DOAS (Setup 1) and by the loss of NO with the chemiluminescence
detector (Setup 2). The latter was preceded by a molybdenum converter to transform HONO and
NO$_2$ to NO, and by an alkaline trap for HONO. Both, trap and converter, had a bypass to allow
sequential measurements and thereby obtaining the concentration of NO$_2$ and HONO separately.
HONO was measured by the LOPAP during some selected experiments (Kleffmann et al., 2002,

152  2006).

**LED-CE-DOAS**

The LED-CE-DOAS instrument (Thalman and Volkamer, 2010) detects NO$_2$ absorption at blue
wavelengths. A high power blue LED light source (420–490 nm) is coupled to a confocal high
finesse optical cavity consisting of two highly reflective mirrors (R = 0.999956) peaking at 460
nm that are placed about 87.5 cm apart (sample path length of 74 cm). The absorption path length
depends on wavelength, and was about ~11 km near peak reflectivity here. A purge flow of dry
nitrogen gas is added to keep the mirrors clean. The light exiting the cavity is projected onto a
quartz optical fiber coupled to a Princeton Instruments Acton SP2156 Czerny-Turner imaging
spectrometer with a PIXIS 400B CCD detector. The mirror reflectivity was calculated by flowing
helium and nitrogen gas, exploiting the difference in the Rayleigh scattering cross sections of both
gases as described in Thalman et al. (2014). The gas exiting the flow tube was directly injected
into the CE-DOAS cavity, and spectra were recorded every 60 seconds, and stored on a computer.
For analysis we use BBCEAS fitting at NO$_2$ concentrations exceeding few ppbv (Washenfelder et
al., 2008) and DOAS least squares fitting methods at lower concentrations (Thalman et al., 2015).
The mirror alignment was monitored online as part of every spectrum by observing the slant
column density of oxygen collision complexes, O$_2$-O$_2$ (O$_4$) (Thalman and Volkamer, 2010, 2013).
The following reference spectra were taken from the literature: NO$_2$ (Vandaele et al., 2002) and
O$_2$-O$_2$ collision complexes (Thalman and Volkamer, 2013b). The detection limit for NO$_2$ was 50-
100 pptv.

**2.1.2. Experimental conditions**

The IC+CA solutions were prepared by adding IC into a 1 M CA solution in 18 MΩ ultra-pure
water to achieve IC to CA molecular ratios between 0.026 to 0.127 in the film. The bulk solutions
for both Setup were prepared by weighing out 384-400 mg of CA in 2 mL of water and adding 4-
20 mg of IC to the solution. The solutions for both setups were freshly prepared for each



experiment and the masses in the film were calculated at 50% RH from the CA hygroscopic growth
factors reported by Zardini et al., 2008 for both setups (for Setup 1: 5-18 mg of IC and 44 mg of
CA, for Setup 2: 1-5 mg of IC and 77 mg of CA). The range of concentrations in the films was
between 0.148 – 0.671 M.
The IC+CA solution coatings were produced by depositing 220-250 µL (Setup 1) and 400 µL
(Setup 2) of the desired solution in a Duran glass tube, which was then dispersed into a thin and
viscous film. The film was dried with a gentle $N_2$ stream humidified to a RH similar to the
experimental RH and room temperature. The film was rolled and turned upside down to deposit a
homogenous film throughout the entire inner surface of the flow tube. The homogeneity of the
film was confirmed by visual inspection. If a bright clear homogenous amorphous film from the
super-cooled solution was not observed, the film was discarded (e.g. observation of a turbid and
cracked crystallized appearance). The carrier gas flows consisted of premixed dry $N_2$ and $O_2$ (a
ratio of 4.5/1 in Setup 1 and a ratio of 2 in Setup 2), and NO controlled by mass flow controllers.
The total flow rates were: 500 mL/min for Setup 1 and 1500 mL/min for Setup 2. In Setup 1, a
dilution flow of 1000 mL/min was added at the end of the flow tube for a total of 1500 mL/min
during experiments when HONO was measured along with $NO_2$. All experiments were conducted
at ambient pressure, leading to gas residence times of 2.1 – 2.4 s (depending on flow tube volume,
for both setups) under laminar flow conditions. The $O_2$ flow rate was varied between 0-110
mL/min to observe the dependence of $O_2$ while keeping the total flow rate constant. A ratio of
4.5:1 of $N_2:O_2$ was maintained if any of the other gas flows were changed (e.g. NO, and/or $NO_2$)
for Setup 1. For Setup 2, a ratio of 2:1 of $N_2:O_2$ was also maintained, except for the $O_2$
concentration dependence studies. The RH was kept constant at 50% RH during most experiments,
and varied between 10-60% RH to study humidity effects of the $HO_2$ radical production. The
concentration of NO was ~1 ppmv (Setup 1) and varied between 100 and 500 ppbv (Setup 2).
Scavenging of $HO_2$ was achieved by the following reaction:
$$NO + HO_2 \rightarrow NO_2 + OH \tag{R1}$$
The lifetime of $HO_2$ is about 5 ms when $2.5 \times 10^{13}$ molecules cm$^{-3}$ of NO are present (Setup 1),
which assures efficient conversion of $HO_2$ molecules into $NO_2$ (k = $8.0 \times 10^{-12}$ cm$^3$ molecule$^{-1}$ s$^{-1}$
at 298 K, Sander et al., 2011). As shown in Fig. S1, 500 ppbv NO, the concentration used in Setup
2, was sufficient to efficiently convert $HO_2$ into $NO_2$, see Sect. 3.1.1. The lifetime of gas phase



HO₂ with respect to loss to the organic film is about 0.1 s, based on a similar formula shown in
Equation S3, where $\gamma = 10^{-3}$ (upper limit by Lakey et al., 2015).  Note that in view of the essentially
diffusion controlled loss of HO₂ to the CWFT and tubing walls, the chosen scheme for determining
the production of HO₂ radicals from the films by fast scavenging with NO is superior to a more
selective detection method, e.g. LIF, which would require passing the HO₂ radicals into a separate
setup with substantial losses.  For selective experiments, the films were exposed to UV irradiation
for over six hours which showed a minor change in the decrease of NO₂ concluding the stability
of the reactivity of the films.
**2.1.3. $J_{IC}$ calculations**
The absorption cross section of IC and the calculated photolysis rate are shown in Fig. S3. The
photolysis frequencies of IC were calculated using a similar procedure as described in
Schwarzenbach et al. (2002). The spectral irradiance in the flow tube system was interpolated to
the surface area of the flow tube to calculate the spectral photon flux density and the absorbed
photon flux:
$$F_a^{IC} = \int_{300}^{420} F \times [1 - 10^{-\sigma_{IC}(\lambda) \times b \times C_{IC}}] d\lambda , \qquad \text{where } F = \frac{F_{FT}(\lambda) \times SA}{N_a \times V_{film}}, \qquad (1)$$
Where $F_a^{IC}$ is the mean absorbed photon flux in Ein L⁻¹ s⁻¹ nm⁻¹ (1 Ein = 3.0 × 10⁵ J per mole of
photons at 400 nm), $F$ is the spectral flux density that reaches the film in the flow tube in moles L⁻
¹ s⁻¹ nm⁻¹,  $b$ is the optical path length taken as the thickness of the film and $C_{IC}$ is the concentration
of IC in the film, and $\sigma_{IC}$ is the IC absorption cross section.  The absorption spectrum of IC in
water was based on the measurements by Kampf et al. (2012), and re-normalized to the peak value
of 10205 ± 2400 M⁻¹ cm⁻¹  at 284 nm (Maxut et al., 2015). $V_{film}$ is the volume of the film calculated
from the deposited mass of CA and the hygroscopic growth factors of CA (Zardini et al., 2008),
$SA$ is the surface area of the flow tube of the film, taken as the geometric area of the inner surface
area of the flow tube in cm², $N_a$ is Avogadro's number in molecules mole⁻¹.  The IC photoexcitation
rate $J_{IC}$ was about $1.0 \times 10^{-3}$ s⁻¹ (upper limit).
We have also attempted to calculate an effective quantum yield for the formation of gas-phase
HO₂ radicals ($\phi_{HO2}$):
$$P_{HO2} = \frac{[NO_2] \times flow}{N_a \times V_{film}} \qquad\qquad \phi_{HO2} = \frac{P_{HO2}}{F_a^{IC}} \qquad (2)$$





235 Where $P_{HO2}$ is the HO$_2$ production rate in mol L$^{-1}$ s$^{-1}$, $F_a^{IC}$ is the calculated mean absorbed photon

236 flux by IC (Eq. 1), $[NO_2]$ is the gas-phase concentration of NO$_2$ in molecules cm$^{-3}$ assuming a 1:1

237 ratio to HO$_2$ conversion, $flow$ is the volumetric gas flow at the temperature in the CWTF and

238 atmospheric pressure in cm$^3$ s$^{-1}$, and $V_{film}$ is in L.

## 2.2. Aerosol flow-reactor experiments

240 A detailed description of the aerosol flow tube (AFT) is reported elsewhere (Monge et al., 2012;

241 Aregahegn et al., 2013), therefore, only some principles are recalled below. The SOA experiments

242 were conducted in a horizontal, cylindrical, Pyrex, aerosol flow reactor (13 cm i. d., 152 cm length)

243 surrounded by seven UV lamps (Philips CLEO, 80W) with a continuous emission spectrum

244 ranging from 300-420 nm (total irradiance of $3.31 \times 10^{16}$ photons cm$^{-2}$ s$^{-1}$). The flow reactor

245 consisted of Teflon stoppers and different flow controllers that maintained the gas/aerosol/UV

246 irradiation contact time between 20-50 minutes. This flow reactor also consisted of an outer jacket

247 that controlled the temperature at 293 ± 2 K by water circulation using a thermostat (Model Huber

248 CC 405).

249 Seed aerosols (50 nm) were produced by nebulizing a solution (at pH 6) containing ammonium

250 sulfate (AS, 0.95 mM) and IC (1.3 mM), size selected by a DMA, and exposed to gas-phase

251 limonene (500 ppbv) in the aerosol flow reactor. The typical aerosol mass loading in the reactor

252 was 2-3 μg cm$^{-3}$, corresponding to ~15000 particles cm$^{-3}$ with a starting diameter of 50 nm. As

253 shown by Aregahegn et al. (in 2013), limonene is an efficient H-donor VOC that forms SOA via

254 reactive uptake to IC containing seed aerosol. Due to the excess of limonene, and low seed aerosol

255 surface are the consumption of limonene was below the detection limit. The aerosol growth was

256 measured by means of an Ultrafine Condensation Particle Counter (UCPC) and a Scanning

257 Mobility Particle Sizer Spectrometer (SMPS; both TSI), and similarly to the CWFT experiment, a

258 flow of gaseous NO (from a 1 ppmv cylinder, Linde) was added to the carrier gas, and its

259 conversion to NO$_2$ monitored by chemiluminescence detector with a detection limit of 0.05 ppbv

260 (ECO PHYSICS CLD 88). Due to the long residence time, the NO$_2$ concentration is affected by

261 its photolysis in the AFT. As discussed below, P$_{HO2}$ was calculated, in this case, from the growth

262 of the particle diameter measured at the exit of the flow tube; the assumption is that growth was

263 due to reactive uptake of limonene only, and that each limonene forms one HO$_2$ radical. At 30

264 ppbv NO, the HO$_2$ radical lifetime is around 2 sec.



### 2.2.1. Experimental conditions

The total flow rate in the aerosol flow reactor was between 400 – 1000 ml/min, ensuring laminar
flow conditions. The RH was varied between 0 – 50%. The RH of particles in the flow reactor
was controlled by saturating the carrier gas via a bubbler containing ultra-pure water (Milli Q, 18
Mohm). The RH in the flow reactor system was varied by changing the gas flow rates to the
bubbler and the temperature of the circulating water jacket of the bubbler. The RH was measured
with a humidity sensor (Meltec UFT 75-AT, Germany) at the exit of the flow reactor. The
concentrations for the flow tube experiments were the following: 30 ppbv of NO and 500 ppbv of
limonene.

### 2.3. Chemicals

The following chemicals were used without further purification for CWFT studies: IC (97%,
Sigma Aldrich), and CA (Sigma Aldrich). For Setup 1, the Duran glass tubes were soaked in a
deconex® cleaning solution overnight, the next day they were rinsed with 18 MΩ water (Milli Q
Element system). These flow tubes were etched with a 5% hydrofluoric acid solution after the
washing procedure and again rinsed with water before any experimental use. The Duran flow
tubes for Setup 2 were not initially etched with any acid but stored in a NaOH solution after
washing and lastly rinsed with water; Setup 2 later confirmed that the treatment of flow tube with
acids affects $P_{HO2}$ by rinsing with HCl and etching with HF solutions.
For the aerosol flow-reactor experiments gas-phase limonene was generated from commercially
available limonene (Aldrich, 97%) by means of a permeation tube. The following chemicals were
used without further purification: IC (97%, Sigma Aldrich) and succinic acid (Sigma Aldrich,
≥99.5%);4-benzoylbenzoic acid (4-BBA, Aldrich 99%) and adipic acid (AA, Aldrich, ≥99.5%)
were used to expand the CWFT studies to other photosensitizers.

### 3. Results and Discussion

### 3.1. Coated-wall flow tube

The following results represent the light dependent formation of $HO_2$ indirectly from
measurements of $NO_2$ production and NO loss, measured with setup 1 and 2, respectively. Figure



2 shows a time series of $NO_2$ measured with setup 1 as a function of UV-A light, which confirms
the light dependent radical production. This particular film had an IC/CA ratio of 0.026 (0.148M
IC and 5.77M CA in the film). An evident increase of $NO_2$ is observed upon UV irradiation,
directly reflecting the light mediated release of $HO_2$, as shown in reaction (R1). The $NO_2$ signal
decrease over time with all seven lamps was a common feature observed in all films; this could be
due to $HO_2$ sinks in the film increasing with time, thus, the system only slowly evolves into a
steady state. A small amount of $NO_2$ (0.5-1.5 ppbv) was observed during experiments that used
only CA in absence of IC; therefore, the data in Fig. 2 and all data reported below have been
corrected for this $NO_2$ background, measured routinely in between experiments. Figure 2 also
indicates a strong correlation with irradiance, which is further discussed in the context of Fig. 4.
Each data point was measured from a freshly prepared coated film in the flow tube. The
uncertainty for experiments was based on the standard deviation of $n$, the number of experiments.
The total uncertainty was ± 6-27% (propagated error for normalization was ± 7–29%) for the IC
mass loading experiments in Setup 1 and up to a factor of two for the light dependence
experiments. The uncertainty in Setup 2 was 10-50%. As discussed earlier, the lifetime of $HO_2$ in
the system was about three orders of magnitude less than the residence time in the flow tube,
therefore suggesting that most, if not all, reacted with NO to produce the observed $NO_2$ (R1).
Theoretically, the system was clean of other oxidants such as $O_3$ (and thus $NO_3$). The uptake of
$NO_2$ in the film was very small to further produce any nitrate radicals, and the photolysis of $NO_2$
in the experiments to produce $O_3$ was insignificant (< 1%). The recombination of NO and $O_3$
contributes a negligible (<0.1%) $NO_2$ source under our experimental conditions. $RO_2$ generation
from the reaction between CA and OH from HONO photolysis was also ruled out since it is
approximated to account for only 1% of the $NO_2$ production if we assume every OH from the
photolysis reacts with CA. To our knowledge, the direct photolysis of CA to produce any $RO_2$
radicals has not been observed. Therefore, we believe that $HO_2$ is the essential oxidant for NO
and refer to the measured $NO_2$ as $HO_2$ formation.
Figure 3 shows that the $HO_2$ production fluxes, in molecules $cm^{-2}$ $min^{-1}$, increased with IC mass
loading. The CA concentration was kept constant, and results are shown as the product between
[IC] × [CA], since we expect that the production rate of $HO_2$ is proportional to the concentration
of IC, at constant illumination, and that of the potential H-donor, CA. For Setup 1, the $HO_2$ fluxes
were measured as $NO_2$ mixing ratios, and calculated using the following equation:





$$Fluxes_{HO_2} = \frac{[NO_2] \times flow}{SA} \qquad (3)$$
the description of these parameters have been previously explained (see Sect. 2.1.3). For Setup 2,
the $HO_2$ flux was calculated similarly, but only about half of the observed NO was considered to
account for the loss of NO via the reaction with OH (see reaction in Supplement R1), meaning that
for each $HO_2$ scavenged two NO molecules were lost. In Figure 3, the data from Setup 1 are
represented by the black squares and the data from Setup 2 are represented by the gray circles.
Setup 1 measurements were taken at about ~50% RH and at room temperature. Setup 2
measurements were taken at 45% RH and at 292 K. Temperature has an effect on the observed
gas-phase $HO_2$ release from the film and thus needs to be accounted for, which is not accounted
for in Fig. 3 but it is described in detail in Sect. 3.1.1.
Figure 4 shows that the $HO_2$ production exhibited a linear dependence on the actinic flux for
various [IC] × [CA] molar products. From Sect. 2.1.3, we estimated an experimental $\phi_{HO2}$ of
about $6 \times 10^{-5}$, reflecting other probable, unknown quenching processes in our system. Figure 4
also shows the formation of HONO from three different IC mass loadings. In all three cases the
HONO:$NO_2$ ratio is < 1, confirming $HO_2$ as a primary product and OH as a secondary product.
Figure 5 shows the dependence of $HO_2$ production observed via the loss of NO (Setup 2) on
relative humidity (0 – 65%). Water partial pressure is an important parameter in the atmosphere
and it seems to also have an important effect on the photochemical reactions studied here. At RH
below ~10%, and at high RH above ~55%, the yield of $HO_2$ radicals decreases. The maximum
$HO_2$ radical production is observed at moderate RH (20 – 55%). This is probably due to a
combination of factors. In particular, at low RH the film may become more viscous reducing
mobility, and thus the energy transfer within the film. This may decrease the $HO_2$ yield as shown
in Fig. 5. The reduced diffusivity of $HO_2$ may also increase the residence time in the film and
facilitate the self-reaction in the bulk phase: The diffusivity of $H_2O$ in citric acid is in the range of
$10^{-7} – 10^{-8}$ cm$^2$s$^{-1}$ at 50% RH. If the $HO_2$ diffusivity is between a factor of 10 and 100 lower than
that of $H_2O$ due to its larger size, $10^{-9}$ cm$^2$s$^{-1}$, the first order loss rate coefficient for diffusion out
of the film, $D/\delta^2$, $\delta$ denoting the film thickness ($4 \times 10^{-4}$ cm), becomes about $k_D = 10^{-2}$ s$^{-1}$. From the
observed $F_{HO2}$, the steady state concentration is then about $F_{HO2}/ k_D /\delta = 4 \times 10^{16}$ cm$^{-3} = 10^{-7}$ M. The
loss rate coefficient due to $HO_2$ self-reaction in the condensed phase ($7.8 \times 10^5$ M$^{-1}$ s$^{-1}$) at this
concentration would become nearly 0.1 s$^{-1}$, somewhat higher than that for diffusional loss. Of



course these estimates carry a high uncertainty, but indicate that at lower humidity diffusivity gets
low enough to effectively reduce the diffusional loss of $HO_2$ to the gas phase and favor its loss by
self-reaction in the condensed phase. The potential presence of condensed phase sinks, such as
$RO_2$, formed from secondary chemistry of oxidized citric acid may add to this uncertainty. At
high RH (> 55%), the amount of water associated with CA dilutes the reactants, and quenching of
the excited IC triplet states gains in relative importance, consistent with findings in other studies
(Stemmler et al., 2006, 2007; Jammoul et al., 2008). The RH effect can decrease the $HO_2$
production by a factor of 3, compared to the plateau of maximum $HO_2$ production between 20 –
55% RH.
Figure 6 shows the dependence of the $HO_2$ production based on the observed NO loss on the $O_2$
mixing ratio (Setup 2). The $HO_2$ production varied by about 20% over the range of conditions
investigated. A marginal decrease below 15% $O_2$ appears to be significant compared to the
maximum $HO_2$ production at ~40% $O_2$, indicating that $O_2$ is needed for $HO_2$ formation. Sufficient
$O_2$ dissolves in the aqueous phase to produce $HO_2$ radicals efficiently at atmospheric $O_2$ mixing
ratios. Above 55% $O_2$ the $HO_2$ production decreased, which is probably due to quenching of
excited IC triplet states by $O_2$. Our results are qualitatively consistent with the observations of
decreasing aerosol growth at high $O_2$ in the autophotocatalytic aerosol growth described in
Aregahegn et al. (2013).
In order to test the possibility for excited IC triplet states to react with $NO_2$ at the surface of the
film, experiments were conducted with $NO_2$. While we did observe that the uptake of $NO_2$ on
irradiated surfaces scaled with light intensity (see Fig. S4) the reactive uptake coefficient of $NO_2$
to produce HONO at the surface is rather small (< 2.5 x $10^{-7}$), corresponding to a $k_w$ of $10^{-3}$ $s^{-1}$ and
thus neither a significant loss of $NO_2$ nor a significant source of HONO. The primary fate of the
nitrogen-containing aromatic alkoxy IC radical under atmospheric conditions is reaction with $O_2$.
However, we have not tested alternative quenching reactions of the triplet state, or other pathways
of the reduced ketyl radical that do not result into formation of HONO.

### 3.1.1 Comparison of data sets

The experimental conditions probed differ in the actinic flux, NO concentration, temperature, and
acidity. Here, we use the dependencies established in Sect. 3.1 to compare results from both setups.
The data from Setup 2 were normalized to conditions of Setup 1. The difference in $J_{NO2}$



corresponds to multiplying results from Setup 2 with a factor of $2.0 \pm 0.1$. $HO_2$ was measured
indirectly by reacting it with NO, and Fig. S1 indicates the minimum NO concentration needed to
efficiently scavenge all gas-phase $HO_2$ is ~460 ppbv of NO, indicating efficient conversion for
Setup 1, and a conversion efficiency of ~0.6 for Setup 2. The data from Setup 2 were multiplied
by $1.66 \pm 0.10$ to normalize for the NO conversion efficiency (Fig. S1), and by an additional factor
$1.25 \pm 0.10$ to match temperatures. We observed some limited variability depending on whether
HF or HCl were used to clean the flow tube prior to experiments. A higher $P_{HO2}$ was observed
when cleaning with HF (Setup 1) compared to storing in NaOH and either rinsing with water or
HCl (Setup 2); this is accounted by multiplying data from Setup 2 with a factor of $1.25 \pm 0.30$.
Notably, the error of the correction for the cleaning procedure that is propagated here is larger than
the correction factor. The effect of the pretreatment of the flow tubes was not systematically
studied, and thus remains a primary uncertainty in the comparison. No further correction was
applied for slight differences in RH. The overall correction factor amounts to $5.2 \pm 1.4$, with the
error reflecting the propagated uncertainty. This explains most the difference in $P_{HO2}$ between both
setups. The normalized results agree within a factor of 2, which is a reasonably good agreement.
**3.1.2 Extension to other photosensitizers**
A limited number of experiments were performed using the CWFT approach, using 4-BBA as a
photosensitizer, in presence of 790 ppbv of gaseous limonene (a possible H-donor) and NO. The
organic thin film contained an organic acid matrix made of 4-BBA with/without adipic acid. Also
in this system a substantial conversion of NO into $NO_2$ was observed. That 4-BBA behaves similar
to the IC system demonstrates that the chemistry discussed above can occur on different excited
carbonyls. It is interesting to note that this photo-induced conversion, and $HO_2$ production, was
observed to be sustained over long times i.e., more than 15 h probably due to the catalytic nature
of the underlying chemical cycles. However, a fraction of the IC did get consumed by photolysis
reactions that do not form the excited triplet state (observed during overnight experiments). The
$HO_2$ flux for the 4-BBA system was estimated to be $2.77 \times 10^{10}$ molecules $cm^{-2}$ $min^{-1}$ making the
same assumption that each $HO_2$ molecule reacts with NO to generate an $NO_2$ molecule. The
calculation is based on Eq. 3, where it depends on the concentration of $NO_2$ as well as the surface
area and residence time.
**3.2. Aerosol Flow Tube**





The aerosol flow tube experiments were conducted similarly to the study by Aregahegn et al.
(2013), i.e., who demonstrated that in the absence of NO and known gas phase oxidants, seed
particles containing IC can initiate SOA growth in presence of a gaseous H-donor (limonene).
Figure 7 shows the results from similar experiments when NO was added to the system. No
conversion of NO to $NO_2$ was observed prior to the injection of limonene into the flow tube. The
presence of a gaseous H-donor and light clearly initiated a series of photochemical processes,
leading to SOA growth and gaseous $NO_2$ production. However, the quantitative interpretation
of these experiments is not straightforward due to efficient radical cycling in the
VOC/$NO_x$/light photochemical system, and the lack of a blank experiment that did not contain
IC as part of the seed particles. Limitations arise from the much longer residence time, which
allows $NO_2$ to be significantly photolyzed. The $J_{NO2}$ was estimated as ~6.75 x $10^{-3}$ $s^{-1}$, and
corresponds to a photolysis lifetime of 2.5 minutes, which is smaller than the actual residence
time in the flow tube (~40 mins). Secondary chemistry can lead, among others, to ozone
production ($O_3$ lifetime at 500 ppbv limonene is ~7 min), and secondary OH radical formation
from the ozonolysis of limonene. Notably, $NO_x$ is not consumed in Fig. 7. The overall effect
of this secondary chemistry is an increased SOA growth compared to an experiment without
added NO (Aregahegn et al., 2013). As a consequence, the $NO_2$ yield cannot be used directly
to assess $P_{HO2}$ in presence of NO.
However, in the absence of NO these secondary processes can largely be avoided, and are
reduced at a level where they cannot be identified (Aregahegn et al., 2013). Under such
conditions, the particle growth rates presumably carry information about the photosensitizer
cycling and subsequent $HO_2$ production. If we assume one molecule of limonene reacts to
produce one $HO_2$, the volume change of aerosols is proportional to the overall number of $HO_2$
produced. For example, a growth of 15,000 particles $cm^{-3}$ from diameter 51.4 nm to 68.5 nm in 40
mins (residence time) is equal to $P_{HO2}$ of $1.67 \times 10^{14}$ molecules $cm^{-2}$ $min^{-1}$. This should be
interpreted as an upper limit for the actual $P_{HO2}$, because water uptake may also be
contributing to the volume growth. However, compared to the CWFT experiments the much
higher surface to volume ratio of nanoparticles is expected to enhance the chemical coupling
of a gas-phase H-donor and the excited IC triplet state at the aerosol surface. This is at least
in part deemed responsible for the two orders of magnitude higher $P_{HO2}$ in the aerosol flow





tube compared to the CWFT experiments. Notably, even if $\phi_{HO2}$ in the aerosol flow tube was
two order of magnitude higher than in the CWFT, it is still significantly smaller than unity.

**Primary HO₂ formation from IC**

One of the main advantages of the CWFT is that it operates at much shorter residence time. From
Setup 1, we derive a $P_{HO2}$ of $1.76 \times 10^{12}$ molecules cm$^{-2}$ min$^{-1}$ for IC/CA = 0.1 and $J_{NO2} = 8 \times 10^{-3}$
$^{3}$ s$^{-1}$. This corresponds to $2.9 \times 10^{4}$ molecules cm$^{-3}$ s$^{-1}$ once normalized by aerosol surface area
($1.18 \times 10^{-6}$ cm$^{2}$ cm$^{-3}$), and $J_{NO2}$ in the aerosol flow tube. Such a primary radical flux is equivalent
to the OH radical production rate resulting from photolysis of ~1 pptv of HONO in the aerosol
flow tube. Conversely, a $P_{HO2}$ of $1.67 \times 10^{14}$ molecules cm$^{-2}$ min$^{-1}$ is equivalent to the OH radical
production rate from ~100 pptv HONO in the aerosol flow tube. We conclude that seed particles
containing IC contribute significantly (equivalent to 1-100 pptv HONO) to the primary HO$_x$ radical
production rate in the aerosol flow tube experiments in the presence of NO (Fig. 7). Primary HO₂
radicals formed from IC containing seed particles react rapidly with NO to form OH radicals under
the conditions shown in Figure 7. The H-donor species is further expected to form primary RO₂
radicals. These primary HO₂ and RO₂ radicals add directly to the conversion of NO into NO₂, and
indirectly by driving secondary NO-to-NO₂ conversion from the RO₂/HO₂ radical chain. The
aerosol flow tube experiments thus qualitatively confirm the results obtained from macroscopic
surfaces, and highlight the potentially important role of surface-to-volume ratio and gaseous H-
donors to enhance the relevance of H-donor photochemistry as sources for HO$_x$/RO$_x$ radicals and
SOA.

**3.3. Proposed mechanism**

A mechanism that can describe the results from the CWFT experiments is shown in Fig. 8. It
follows the mechanism first proposed by Canonica et al., in 1995. The primary product in our
system is the HO₂ radical, which forms from the reaction between a nitrogen-containing aromatic
alkoxy IC radical and a ground state oxygen molecule, recycling the IC molecule. The aromatic
alkoxy radicals form from the excited triplet state of IC via transfer of an H atom from an H-donor
(in our case likely to be CA, or the CA/H₂O matrix). While a fraction of the IC will get consumed
by photolysis reactions that do not form the excited triplet state (see Sect 3.1.2.), IC is also
continuously produced from multiphase reactions, e.g., of glyoxal (Yu et al., 2011; Kampf et al.,
2012; Maxut et al., 2015). Another conclusion is that OH is a secondary product. If OH was a first





generation product we would have expected HONO:NO$_2$ ratios larger than 1:1. A smaller ratio
was observed, as shown in Fig. 4, indicating that there was no direct evidence for primary
formation of OH radicals. Interestingly, the H-donor species becomes activated as a result of H-
abstraction, and can react further to produce organic peroxy radicals, as evidenced by the aerosol
flow tube results.

**4. Atmospheric relevance**
The atmospheric relevance of our findings consists of the possible effect of heterogeneous radical
sources to modify atmospheric HO$_2$ radical concentrations, and facilitate aerosol growth and
ageing by adding a radical source within aerosol particles. The production of gas-phase HO$_2$ from
IC photosensitized heterogeneous chemistry is a possible source of gas-phase HO$_2$ radicals in
ambient air. In order to estimate the possible relevance for HO$_2$ radical concentrations in urban air,
we assume P$_{HO2}$ of $2 \times 10^{12}$ molecules cm$^{-2}$ min$^{-1}$ (IC/CA = 0.1, Setup 1) as a lower limit, and 2
$\times 10^{14}$ molecules cm$^{-2}$ min$^{-1}$ (IC/AS = 0.1, aerosol flow tube) as an upper limit, and typical
conditions in Mexico City (i.e., $J_{NO2} = 8 \times 10^{-3}$ s$^{-1}$ at noontime in Mexico City, aerosol surface
area = 15 cm$^2$ m$^{-3}$; Volkamer et al., 2007). The normalized P$_{HO2}$ during noon time in Mexico City
ranges from $2 \times 10^5$ to $2 \times 10^7$ molecules cm$^{-3}$ s$^{-1}$. This corresponds to a rate of new HO$_2$ radical
production of 4 to 400 pptv HONO around solar noon in Mexico City (Li et al., 2010), where other
radical sources produce about $5.9 \times 10^7$ molecules cm$^{-3}$ s$^{-1}$ at solar noon (Volkamer et al., 2010).
The upper range value suggests that aerosol surfaces can be a significant source of gas-phase HO$_x$
in places like Mexico City. However, the IC molar ratios used here are likely an upper limit
compared to ambient aerosols, yet, in principle other brown carbon molecules (i.e. HULIS and/or
other imidazole derivatives) may form additional gas-phase HO$_2$. The heterogeneous HO$_2$ radical
source could further be relatively more important in unpolluted regions under biogenic influences,
where gas-phase radical production rates are lower. A more comprehensive characterization of the
heterogeneous HO$_2$ source effect on gas-phase HO$_2$ radical concentrations hence deserves further
investigation.
OH radical uptake from the gas-phase is a primary OH source in aerosols (Ervens and Volkamer,
2010). Assuming a gas-phase OH concentration of $10^6$ molecules cm$^{-3}$, 15 cm$^2$ m$^{-3}$ aerosol surface





area, and $\gamma_{OH}$ of unity, the rate of OH uptake is approximately $2.3 \times 10^5$ molecules cm$^{-3}$ s$^{-1}$. The
above estimated $P_{HO2}$ is a result from H-transfer to form organic peroxy radicals which is
comparable to the rate of OH uptake. The two similar estimates of $HO_x$ suggest that IC is a
significant source of radicals in the condensed phase of particles. This is a lower limit due to the
unknown radical losses of $HO_x$ to the condensed phase, which hold potential to leverage the HOx
source by up to a factor 10,000 if limited by the IC excitation rate. The unknown amount of $HO_2$
that remains in the condensed-phase is a further source of OH in the condensed-phase that can start
Fenton reactions (if iron is present) or other oxidizing pathways that can further age the aerosol.
These results show that IC, and other aromatic carbonyl photosensitizers, are likely a relevant
radical source in aerosol particles. Photo-induced radical generation in condensed phases is
currently not represented in atmospheric models that describe aerosol ageing, and warrant further
study.

**5. Conclusion**
Three different experimental setups consistently show that $HO_2$ radicals are produced from the
photochemistry of IC in a $CA+H_2O$ matrix and in seed aerosols containing ammonium sulfate (in
presence of a gas-phase H-donor, limonene). The linear correlations of $P_{HO2}$ (with [IC]/[CA] and
irradiation) yielded maximum $P_{HO2}$ under atmospherically relevant irradiation, $O_2$ and RH, but
also revealed a complex role of film viscosity, and possibly acid effects. If the H-donor species is
in the condensed phase, significant amounts of $HO_2$ reach the gas-phase only for moderately high
RH (~25 – 55% RH) that facilitates H-transfer, and allows molecules (IC, $HO_2$) to move freely
towards the surface of the film. When the film was too dry this mobility is inhibited due to enhance
viscosity and significantly decreases the $P_{HO2}$. At RH and $O_2$ higher than 55%, we observe a
decrease in $P_{HO2}$ probably due to dilution by water and competing quenching reactions in the film.
If the H-donor species is in the gas-phase, significant $HO_2$ production is also observed under dry
conditions. The primary fate of the IC•-OH radical at the surface is reaction with $O_2$ to form $HO_2$.
$NO_2$ reactions do not appear to form HONO at the surface. Our results suggest that the radical
source from photosensitizers such as IC can help jump-start photochemistry of VOCs. The effect
on the gas-phase $HO_2$ radical concentration increases for higher surface to volume ratio of aerosols,
and in the presence of gas-phase H-donors. The autophotocatalytic growth of aerosols containing





photosensitizers via H-donor chemistry is a SOA source also in the presence of NO, and adds
oxidative capacity inside aerosol particles.  Further research on other types of H-donors and
photosensitizers is necessary to compare different $P_{HO2}$ and rates of aerosol growth from reactive
uptake of VOC that could potentially have a significant atmospheric relevance for SOA formation
and heterogeneous aerosol ageing.
**Author contributions**
M.A. and R.V. designed the experiments at PSI; C.G. and B.N. those at IRCELYON. L.G.P.,
P.C.A., and K.Z.A. conducted the measurements, analyzed data, and contributed equally to this
work.  S.S.S., T.B.R. helped during the experiments, and all co-authors contributed to the data
interpretation.  L.G.P. and R.V. prepared the manuscript with contributions from all co-authors.

**Acknowledgements**
This work was supported by the US National Science Foundation under awards ATM-847793 and
AGS-1452317. M.A. and C.G. appreciate the contribution by the EU project PEGASOS (EU–FP7
project under grant agreement no. 265307).  M.A. appreciates the Swiss National Science
Foundation (grant 130179).



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





**715**  **Figures**

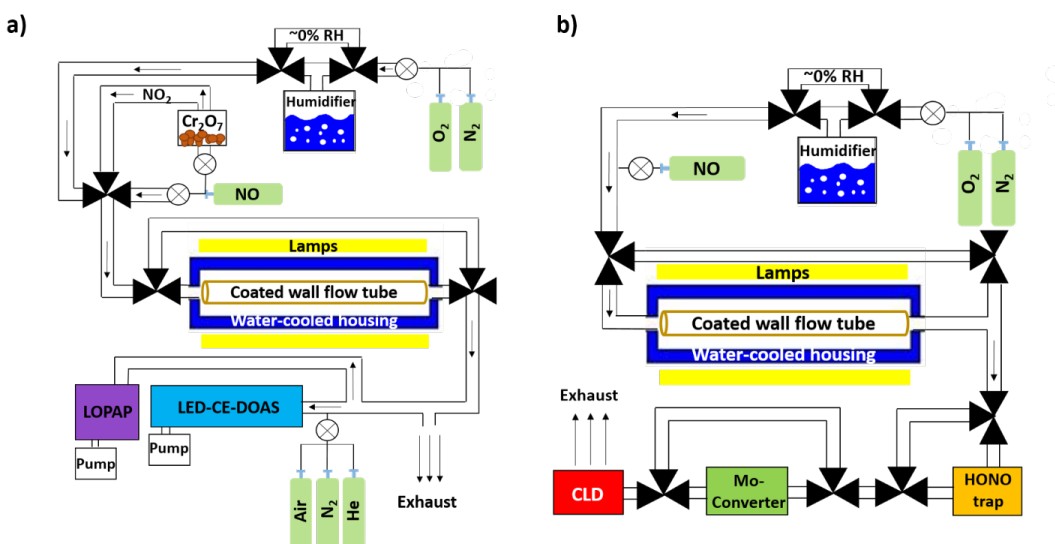


Figure 1. Sketch of the photochemical flow tube reactor setups at PSI for a) Setup 1 in 2013
measuring NO$_2$ generation and b) for Setup 2 in 2014 measuring NO loss.




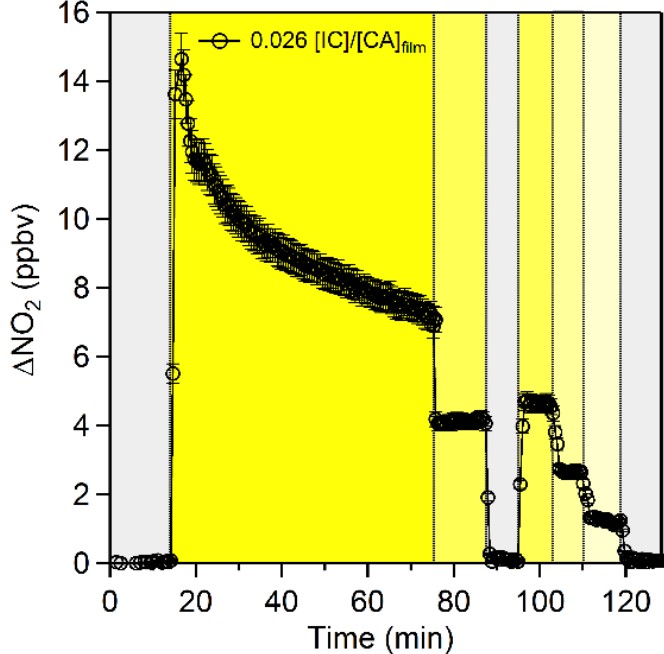



Figure 2. NO₂ profile for a 0.025M IC bulk solution, whose concentration increases to ~0.2 M of
IC in the film due to the citric acid hygroscopic properties. The gray shaded areas indicate
periods where NO was exposed in the dark. The yellow shaded areas indicates the period of
irradiation; the decrease in the intensity of yellow represents the decrease in irradiance (2.26 x
$10^{16}$, 1.47 x $10^{16}$, 1.14 x $10^{16}$, and 3.94 x $10^{15}$ photons cm$^{-2}$ s$^{-1}$, for seven, five, three and one
lamp, respectively). This timeseries clearly indicates the light dependence production of HO₂
radicals from the photosensitization of IC in a CA film.






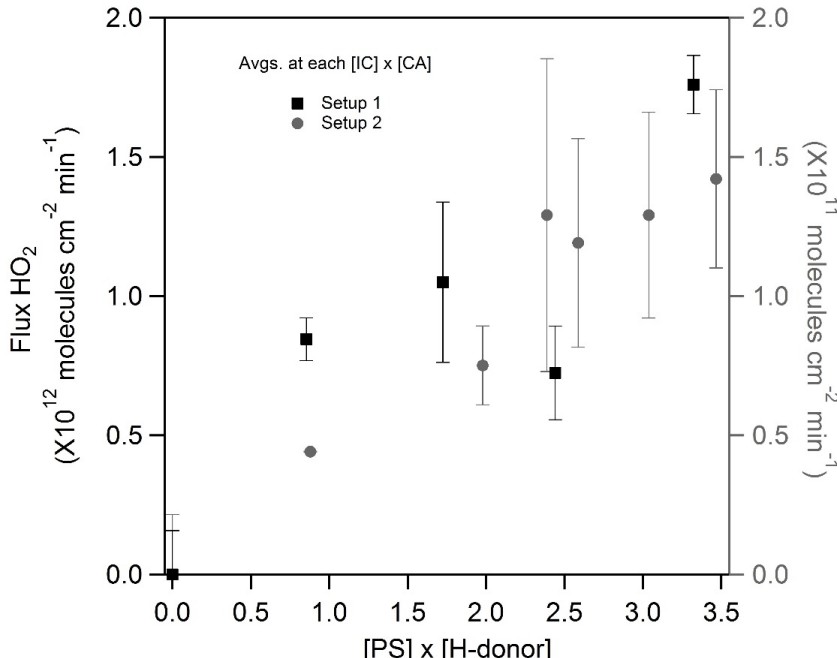


Figure 3. A linear correlation of $HO_2$ as a function of IC concentration. The left y-axis
represents the values for Setup 1, while the right y-axis represents the values for Setup 2, (an
order of magnitude difference for both scales). The Setup 2 data falls between a factor of 2 and 3
from Setup 1 after accounting for differences between Setup 1 and 2, see Sect. 3.1.1.





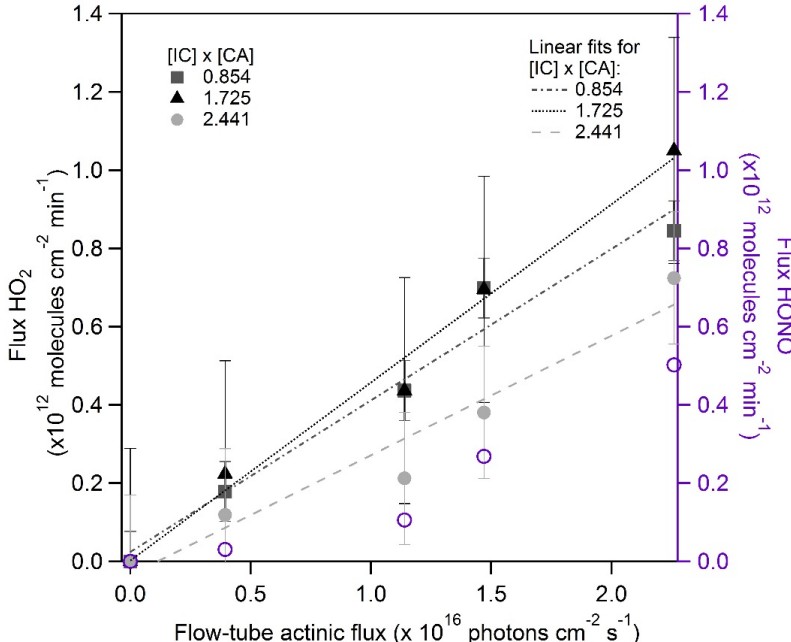



Figure 4. HO$_2$ fluxes (molecules cm$^{-2}$ min$^{-1}$) as a function of actinic flux for a 300-420 nm range.
The data is plotted as a concentration product of [IC] x [CA] (shown in the legend) which shows
the photochemical reaction between IC and CA in H$_2$O matrix and gaseous NO.  HONO for
2.441 ([IC] x [CA]) is plotted on the right axis, showing a ratio of HONO:NO$_2$ < 1, which
suggests OH as a secondary product.





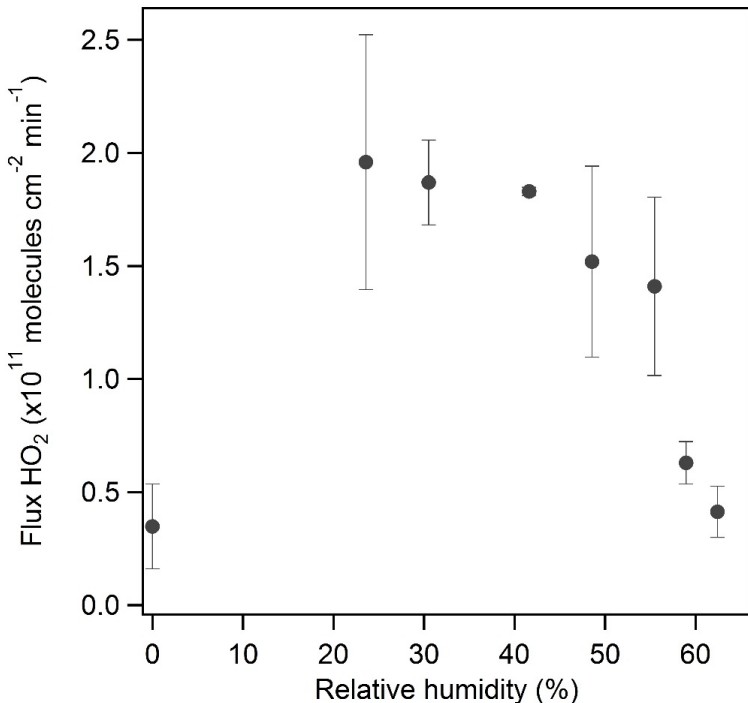



Figure 5. The NO loss normalized to the film surface area as a function of relative humidity.



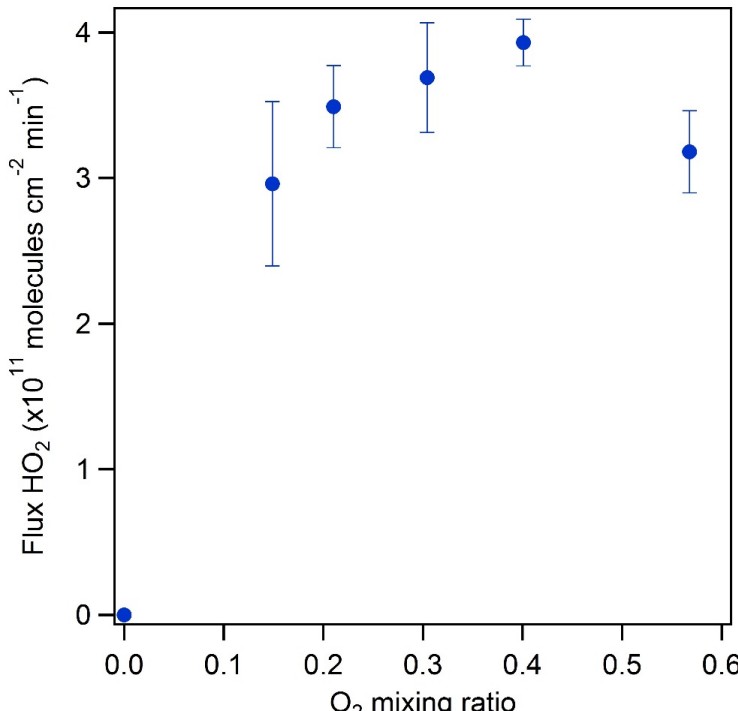



Figure 6. Measured loss of NO above a film composed of IC and CA normalized of the film
surface area as a function of the $O_2$ mixing ratio.




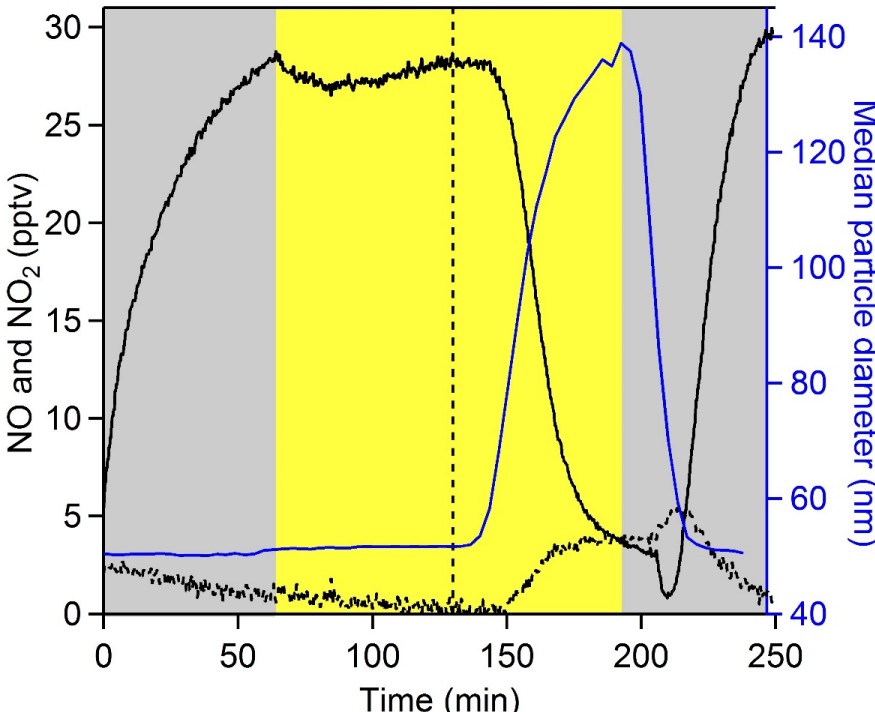


Figure 7. Aerosol flow tube experiments show rapid conversion of NO (solid black line) into
NO$_2$ (dashed black line)   only after the time when limonene (gaseous H-donor) is added into the
flowtube (vertical dashed line). The gray shaded areas represent experiment in the dark, and the
yellow shaded area represents the experiment under light exposure.  The blue line represents the
growth of aerosols, right axis.

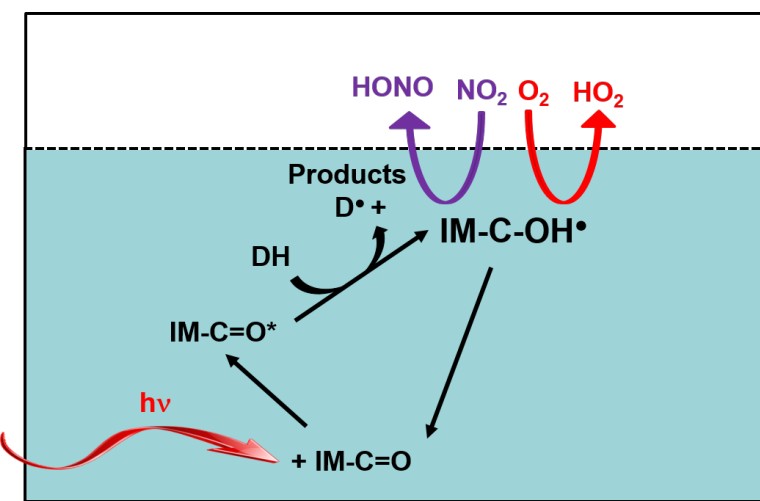



Figure 8. Proposed mechanism, modified and expanded to photosensitization of IC based on
Canonica et al. (1995), George et al. (2005) and Aregahegn et al. (2013). The reaction in the
white square represents the gas-phase, and the blue square represents the aqueous phase. DH is
an H-donor (e.g. CA, another IC, H2O+CA matrix to be determined from flash photolysis).