# Peer review of "Heterogeneous photochemistry of imidazole-2- carboxaldehyde: HO₂ radical formation and aerosol growth"

_Atmospheric Chemistry and Physics, 2016_

## Referee Comment (RC1) · Anonymous Referee #1 · 16 Mar 2016

This is a nice manuscript that should be of great interest to the atmospheric community. HO2 production from aerosols has not been widely investigated; this work suggests that it should not be ignored. I recommend that this manuscript be published following some minor revisions and clarifications.

Major comments:

Is the reaction occurring only at the surface, or also in the bulk? Did PHO2 depend at all on film thickness? If only the surface reaction matters, then bulk phase diffusion (and reactions) can be ignored. If production in the bulk is important, I would expect HO2 production to depend on film thickness. Did it?

What is the expected extent of light attenuation throughout the film? If light is attenuated by the film, photolysis in CWFT experiments would occur primarily at the glass-film interface, and any HO2 formed would have to diffuse to the surface before being released to the gas phase. Conversely, photolysis in the aerosol flow tube would occur primarily at the aerosol-air interface, and no diffusion would be required prior to HO2 desorption to the gas phase. Could this explain the greater gas-phase HO2 production in the aerosol flowtube compared to in the CWFT (either in addition to or in place of increased surface area of the aerosols)?

In the Conclusion the authors state that HO2 production is reduced at dry film surfaces due to increased viscosity (and therefore decreased IC and HO2 mobility) within the film. Is it possible that the reactions occur at the surface and are enhanced by the presence of water? At what RH is a monolayer of water expected to exist at the film surface? Could ionized citric acid behave differently than molecular CA?

Minor Comments: Equation 1 uses 300 nm as a lower limit. Is this appropriate? Is light at that wavelength absorbed by the jacketed flow tube? Even a small change in the wavelength limits could change the calculated photon flux significantly.

Equation 1 also bases the absorption cross section of IC on that measured in aqueous solution. Is the absorbance of IC in a solid film expected to be the same as that in aqueous solution?

There is only one data point at O2 levels greater than 55% shown in Figure 6. I would feel more comfortable with the stated conclusion that HO2 production decreases above 55% O2 with more measurements at higher O2 fractions.

p. 4 lines 103-106: "... that in absence of other known radical sources confirm that HO2 production from aerosols can start photochemistry." I don't understand what this sentence means. Doesn't photochemistry form the HO2?

p.7 line 182-183: What do you mean by a "thin" film? Be a bit more quantitative.

[Figure]

p. 11 line 303: "Each data point was measured from a freshly prepared coated film in the flow tube." What is meant by "each data point"? Figure 2 is a time series; presumably each data point was acquired during the same experiment (using the same film).

p. 13 line 365: In describing Figure 6, the authors say that "A marginal decrease below 15% O2..." Marginal might not be the most appropriate term, since HO2 production decreases to zero in the absence of O2.

p. 18 line 508 -510: OH does not start Fenton reactions. Please reword.

Figure 4: What are the open circles in the plot?

―――――――――――――――――――――

---

## Referee Comment (RC2) · Anonymous Referee #2 · 20 Mar 2016

The work by González Palacios et al. provides indirect evidence for the formation of HO2 during photosensitized reactions of aerosols and thin surface films containing an imidazole and a H-atom donor (e.g., citric acid or limonene). This is a complicated system. The results are of high quality and appropriate methods and techniques were applied to the problem. I recommend publishing after the following points are addressed.

It was not clear how the data support that this reaction is autocatalytic? Are not autocatalytic processes characterized by a logistic product profile, whereas the NO2 vs time profile shown in Figure 2 show NO2 formed decreasing between 20 and 75 minutes. Is there any indication that this will level off at some significant steady state at

later reaction times? On line 296-299 the authors state that the system only slowly evolves into a steady state; however how long does that take? If it does not reach steady-state, and the system ceases to convert NO into NO2 over time, then this is not an autocatalytic system.

The paragraph starting on Line 70 explains the motivation behind studying imidazole photochemistry. As stated by the authors, imidazole is thought to be generated from the reaction of glyoxal with ammonium and amines. The authors only discuss imidazole in SOA has been strictly in laboratory settings (see references cited), which may or may not be reflective of the real environment. I feel this section should include current views (for and against) on the importance of imidazole in atmospheric samples. The recent article by Teich et al. (ES&T 2016, 50, 1166) may be of help here. Such a discussion will help to better convey the atmospheric significance of this study.

Did the authors determine the concentration of nitrate present in their coatings during or following their reactions? When such high levels of NOx, it may be possible that some nitrate could be deposited to the surface. In such a case, photolysis of nitrate could release NO2 into the gas phase. Also, what was the pH of the films coating the glass walls? If citric acid was used this could have lowered the coating pH, in which case one must consider acid-base chemistry as well. This could impact organic photochemical reaction intermediates and also the yield of HO2, NO2 (via nitrate photolysis) and HONO generated in the system.

In the case of the aerosol flow reactor, citric acid was not used; Limonene was used instead, which would not acidify the particle. This represents a major difference between the CWFT and aerosol flowtube measurements that was not discussed. I would be interested in seeing if the authors think this difference could explain why NOx consumption in the aerosol system was so different than during the CWFT experiments.

line 40. The phrase, "...implications consist in a..." is awkwardly phrased. Perhaps revise to say rather: "Our results indicate a potentially relevant contribution..."

line 58. Include "concentrations" after OH2

lines 83-84: Paragraph too long. I suggest inserting a paragraph break between these lines

lines 128-134: It appears that J-values for NO2 photolysis were calculated using clean flow tubes (i.e., in the absence of a coating). Therefore, one is assuming that the J-values for the clean tube are the same as those for a coated tube. Does the IC coating on the coated-wall flow tube attenuate the light enough to invalidate this assumption? If the IC coating does attenuate the light transmitted through the flowtube, how could this bias interpretation of the results?

line 253 (and other places where the term "H-donor" is used): The authors might want to clarify that the VOCs are H-atom donors, rather than proton donors as in the case of a Bronsted acid.

lines 314-318: It is not clear to me: Do the authors think that the NO to NO2 conversion by HO2 is occurring in the organic surface film or in the gas phase? How can they be sure?

lines 319-323: The concentration of citric acid [CA] is in such excess relative to the imidazole. Could it not be assumed that [CA] is constant over the course of the experiments? In that case, why not just plot [HO2] vs [IC] instead of vs [IC]x[CA].

lines 344-347: The authors discuss the role of coating viscosity on photochemistry. A recent article by Hinks et al. (PCCP, 2015, DOI: 10.1039/c5cp05226b) discusses this effect nicely and should be cited here.

line 361: Cannot also H2O photophysically quench the triplet excited state? Is this important for this system?

lines 403-406: The authors mention that the photosensitizer 4-BBA behaves similar to IC. I would like to see this data included in the SI.

[Figure]

lines 426: With respect to ozone formation in the flowtube and aerosol systems, did the authors measure ozone and can they confirm that it was not observed in the system? Lifetimes of ozone were indicated for these experiments, do they account for heterogeneous loss as well?

Figure 8: Focusing on the "IM-C-OH(dot)" intermediate, should not the dot be centered on the carbon and not the alcohol H?

[Figure]

---

## Author Comment (AC1) · 6 Jun 2016

We thank both of the reviewers for their helpful comments and suggestions. In the following we respond to all of the reviewer comments. The Reviewer Comment is first copied using regular text in black, followed by our response using italic font in blue. A copy of the text that we have changed in the manuscript is also added (in green) to facilitate a simultaneous consideration of the reviewers' comments and our replies where appropriate.

**Anonymous Referee #1**

This is a nice manuscript that should be of great interest to the atmospheric community. $HO_2$ production from aerosols has not been widely investigated; this work suggests that it should not be ignored. I recommend that this manuscript be published following some minor revisions and clarifications.

Major comments:

Is the reaction occurring only at the surface, or also in the bulk? Did $P_{HO_2}$ depend at all on film thickness? If only the surface reaction matters, then bulk phase diffusion (and reactions) can be ignored. If production in the bulk is important, I would expect HO2 production to depend on film thickness. Did it?

*Response: The reaction occurs both in the bulk and at the surface of the studied films, as $HO_2$ production was observed for both solid and liquid (viscous) films. In the latter case, an experiment performed at PSI by PCA where IC:CA ratio was kept constant shows the classical behavior of reaction governed by reaction and diffusion (see figure below). At low thicknesses, $P_{HO_2}$ increases linearly with thickness, but saturates at higher thickness above 2-3 μm. Therefore, $P_{HO_2}$ represents clearly $HO_2$ production throughout the top few μm. HO2 produced further below is likely lost by self-reaction in the bulk.*

[Figure]

*We have added this Figure in the SI text of the revised manuscript, and added some discussion in Section 3.1.*

What is the expected extent of light attenuation throughout the film? If light is attenuated by the film, photolysis in CWFT experiments would occur primarily at the glass-film interface, and any $HO_2$ formed would have to diffuse to the surface before being released to the gas phase. Conversely, photolysis in the aerosol flow tube would occur primarily at the aerosol-air interface, and no diffusion would be required prior to $HO_2$ desorption to the gas phase. Could this explain the greater gas-phase $HO_2$ production in the aerosol flowtube compared to in the CWFT (either in addition to or in place of increased surface area of the aerosols)?

*Response: The glass-film interface question can be answered by the response above. The films under study were about 3-4 μm; the above figure shows that these values are too thick for the glass-film interface to play a role in the $HO_2$ production observed.*

*For the light attenuation, we base our response on Figure S2. Our $NO_2$ actinometry studies show that there is not a significant decrease of photolysis to the gaseous $NO_2$ in the presence of a citric acid film (these studies were not performed in the presence of IC though). All photons need to penetrate the aqueous film before they can photolyze $NO_2$ in our setup, and if the film attenuated the light, there should be a significant decrease to the $JNO_2$ value when the film is present.*

*Notably, the IC is an optically thin absorber in our films. The optical density (O.D.) of IC in the film at 300 nm is 0.058, at 320 nm is 0.0095 and at 330 nm is 0.0019 (following Eq. O.D. = c × σ × l, where c is 0.300 M of IC in the film, the σ at 300 nm is $9.2 \times 10^{-18} cm^2$ molecule$^{-1}$, the σ at 320 nm is $1.5 \times 10^{-18} cm^2$ molecule$^{-1}$, and the σ at 330 nm is $3.0 \times 10^{-19} cm^2$ molecule$^{-1}$ and l is 3.81 μm. Since the peak of the JIC spectrum is near 330 nm (Fig. S3), the effect of IC inside the film on the JIC is insignificant.*

*We believe that the higher production of $HO_2$ in aerosols (no diffusion needs to be accounted) is due to the potential higher reaction rate coefficient of limonene with the triplet state of IC compared to the reaction rate coefficient of citric acid. We assume this since in the AFT experiments, the concentration limonene that is exposed to the excited triplet state of IC is much lower than in the CWFT experiments.*

In the Conclusion the authors state that $HO_2$ production is reduced at dry film surfaces due to increased viscosity (and therefore decreased IC and $HO_2$ mobility) within the film. Is it possible that the reactions occur at the surface and are enhanced by the presence of water? At what RH is a monolayer of water expected to exist at the film surface? Could ionized citric acid behave differently than molecular CA?

*Response: Water seems to play a role when the H-donor is in the aqueous phase (compared to the aerosol studies where the H-donor is in the gas-phase, and low RH did not seem to have a significant effect on $P_{HO2}$ as it did in the CWFT studies). Figure 5 further suggests a complex role of water. According to Zardini et al. (2008) pure citric acid solution does not effflorescence; therefore, the film remained as a homogeneous aqueous solution under all RH conditions, and*

*thus the conclusion that the lower P$_{HO2}$ is reduced due to lower diffusivity at low RH is well justified.*

*At the high citric acid activities of this study, acidity is very low, and citric acid predominantly is in its non-dissociated form. pH may have an effect on the absorbance of IC and thus probably on the P$_{HO2}$, but this was not investigated.*

*In the revised manuscript we have added the following in line 525 after P$_{HO2}$:* "Zardini et al. (2008) demonstrates that pure citric acid does not efflorescence, this suggests, that the film remained as a homogenous aqueous solution under all RH conditions. This supports our conclusion that there is lower diffusivity at low RHs since the IC/CA reaction is favored by a certain amount of water molecules present, the range which has been previously stated."

Minor Comments:

Equation 1 uses 300 nm as a lower limit. Is this appropriate? Is light at that wavelength absorbed by the jacketed flow tube? Even a small change in the wavelength limits could change the calculated photon flux significantly.

*Response: The limit at 300 nm is appropriate based on a limitation from the emission of the fluorescence lamps used (the emission below 300 nm is significantly low) and the material of the flow tube (DURAN glass, see http://www.duran-group.com/en/about-duran/duran-properties/optical-properties-of-duran.html). Since the peak of the JIC is near 330nm, and rapidly decreasing at shorter wavelengths (Fig. S3), changing the lower wavelength limit is not expected to change JIC more than the reported uncertainty in JIC.*

Equation 1 also bases the absorption cross section of IC on that measured in aqueous solution. Is the absorbance of IC in a solid film expected to be the same as that in aqueous solution?

*Response: Our films are not solid, but an aqueous solution of IC and CA. Nevertheless, matrix effects on the absorbance would deserve further investigation beyond the scope of the present study; we have allowed ample uncertainty on the calculated JIC to take into account such caveats.*

There is only one data point at O2 levels greater than 55% shown in Figure 6. I would feel more comfortable with the stated conclusion that HO2 production decreases above 55% O2 with more measurements at higher O2 fractions.

*Response: Lines 368-371, have been changed to:* "We assume that at 55% O$_2$, the quenching of excited triplet states by O$_2$ has an effect on HO$_2$ production. This effect may decrease HO$_2$ production based on our results being qualitatively consistent with the observations of decreasing aerosol growth at high O$_2$ in the autophotocatalytic aerosol growth described in Aregahegn et al. (2013). However, the experimental focus of this study was based on atmospheric O$_2$ mixing ratios and thus we cannot conclude about the HO$_2$ production at high O$_2$ mixing ratios."

p. 4 lines 103-106: "... that in absence of other known radical sources confirm that HO2 production from aerosols can start photochemistry." I don't understand what this sentence means. Doesn't photochemistry form the HO2?

*Response: The motivation for this sentence comes from the overarching role of gas phase radical sources in atmospheric chemistry. We have modified the sentence. It now reads: "Section 2.2 describe aerosol flow tube experiments that confirm the photochemical production of $HO_2$ radicals in the absence of other known gas-phase radical sources."*

p.7 line 182-183: What do you mean by a "thin" film? Be a bit more quantitative.

*Response: We have changed the sentence to: "…which was then dispersed into a thin $(3-4 \mu m)$ and viscous film." We have also changed line 180 to: "The range of concentrations in the films was between $0.148-0.671$ M of IC and $5.29-6.68$ M of CA."*

p. 11 line 303: "Each data point was measured from a freshly prepared coated film in the flow tube." What is meant by "each data point"? Figure 2 is a time series; presumably each data point was acquired during the same experiment (using the same film).

*Response: We have changed the sentence to: "For irradiation, humidity and oxygen dependence experiments, each data point represents a separate experiment using a freshly prepared coated film in the flow tube."*

p. 13 line 365: In describing Figure 6, the authors say that "A marginal decrease below 15% O2..." Marginal might not be the most appropriate term, since HO2 production decreases to zero in the absence of O2.

*Response: We have removed 'marginal' here.*

p. 18 line 508 -510: OH does not start Fenton reactions. Please reword.

*Response: We have modified the sentence to read: "The unknown amount of $HO_2$ that remains in the condensed phase is a further source of OH in the same phase; this OH, in the presence of reduced metals, can trigger a cycle of Fenton reactions or other oxidizing pathways that can further age the aerosol."*

Figure 4: What are the open circles in the plot?

*Response: We have added a sentence in line 743: "The solid symbols represent the flux of $HO_2$ and the open circles represent the flux of HONO."*

**Anonymous Referee #2**

The work by González Palacios et al. provides indirect evidence for the formation of HO2 during photosensitized reactions of aerosols and thin surface films containing an imidazole and a H-atom donor (e.g., citric acid or limonene). This is a complicated system. The results are of high quality and appropriate methods and techniques were applied to the problem. I recommend publishing after the following points are addressed.

It was not clear how the data support that this reaction is autocatalytic? Are not autocatalytic processes characterized by a logistic product profile, whereas the NO2 vs time profile shown in Figure 2 show NO2 formed decreasing between 20 and 75 minutes. Is there any indication that this will level off at some significant steady state at later reaction times? On line 296-299 the authors state that the system only slowly evolves into a steady state; however how long does that take? If it does not reach steady-state, and the system ceases to convert NO into NO2 over time, then this is not an autocatalytic system.

*Response: We have substituted the term 'autophotocatalytic' by 'photocatalytic'. We agree that our data do not allow for a firm conclusion about the autocatalytic nature of the mechanism. The initial autophotocatalytic statement applies to aerosol growth, not the CWFT films, and was based on previous studies (Aregahegn et al., 2013; Rossignol et al., 2014). We ran experiments in the CWFT overnight with a single film ( > 10 hrs). Consistently the peak NO₂ production is observed shortly upon turning lamps ON; NO₂ reach sort of a plateau within the first hour, and in the overnight experiments NO₂ formation was always observed. While the 10hr integral HO₂ production rate is smaller than the added amount of IC, the 10hr integral photoexcitation rate of IC in the film is larger than the added amount of IC. Hence, the fact that NO2 formation is observed in the overnight experiments supports the photocatalytic claim. Notably, there is nothing in our data that would contradict the earlier conclusions in previous studies (Aregahegn et al., 2013; Rossignol et al., 2014).*

The paragraph starting on Line 70 explains the motivation behind studying imidazole photochemistry. As stated by the authors, imidazole is thought to be generated from the reaction of glyoxal with ammonium and amines. The authors only discuss imidazole in SOA has been strictly in laboratory settings (see references cited), which may or may not be reflective of the real environment. I feel this section should include current views (for and against) on the importance of imidazole in atmospheric samples. The recent article by Teich et al. (ES&T 2016, 50, 1166) may be of help here. Such a discussion will help to better convey the atmospheric significance of this study.

*Response: We have added discussion about field measurements of IC, including the results presented in Teich et al., 2016 in the revised manuscript.*

*The following text was added: "*Field measurements of imidazoles are generally sparse, yet, recently Teich et al. (2016) identified five imidazoles (1-butylimidazole, 1-ethylimidazole, 2-ethylimidazole, IC and 4(5)-methylimidazole) in ambient aerosols in concentrations ranging

from 0.2 to 14 ng/m$^3$. IC, the molecule under study in this article, was measured in its hydrated form in ambient aerosols in three urban areas with signs of air pollution and biomass burning (Leipzig, Germany, Wuqing and Xianghe, China).  The observed quantities of hydrated IC ranged from 0.9 to 3.2 ng/m.  The authors claim that these values could be a lower limit due to high losses of IC during sample preparation indicated by low recovery from standard solutions. This suggests that IC and other imidazole derivatives are present in areas with high pollution and biomass burning.  Field measurements in Cyprus during the CYPHEX campaign in 2014 detected IC and bis-imidazole in ambient aerosol samples (Jakob et al. 2015). The IC diurnal cycles showed the highest concentrations at night (0.02 – 0.115 ng/m$^3$), and lower concentrations during the day, suggesting that ambient concentrations of IC in aerosols are a balance between photochemical sources and sinks. While imidazoles seem to be widespread in polluted and remote areas, the atmospheric implications of IC, and possibly other photosensitizers related to brown carbon light absorption as radical sources in ambient aerosols deserve further study."*

Did the authors determine the concentration of nitrate present in their coatings during or following their reactions? When such high levels of NOx, it may be possible that some nitrate could be deposited to the surface. In such a case, photolysis of nitrate could release NO2 into the gas phase. Also, what was the pH of the films coating the glass walls? If citric acid was used this could have lowered the coating pH, in which case one must consider acid-base chemistry as well. This could impact organic photochemical reaction intermediates and also the yield of HO2, NO2 (via nitrate photolysis) and HONO generated in the system.

*Response: We did not measure nitrate in either systems. However, this response follows Response #3 to Reviewer #1 (please see above).  As stated above, the high CA concentrations in the studied films have a very low pH (not measured).  According to studies from Laskin et al. (2014) in a citric acid-nitrate system HNO3 is in the gas-phase at low pH; this depletes nitrates by 45 – 55%, and lowers the chance of any photolyzed nitrate to contribute to the NO$_2$ observed.*

*We have added the following statement in the Conclusion section of the manuscript:* "A systematic study of the effect of pH on the IC and CA absorption cross-sections, and the product yields from the IC photochemistry is desirable."*

In the case of the aerosol flow reactor, citric acid was not used; Limonene was used instead, which would not acidify the particle. This represents a major difference between the CWFT and aerosol flowtube measurements that was not discussed. I would be interested in seeing if the authors think this difference could explain why NOx consumption in the aerosol system was so different than during the CWFT experiments.

*Response: The higher NO$_x$ consumption in the AFT is probably the result of vigorous gas-phase secondary chemistry, possibly involving NO$_3$ radicals, and the longer reaction times in the AFT experiments; HNO$_3$ formation is further expected to result into significant acidification in the AFT experiments, but not in the CWFT experiments. The NOx consumption was not the focus of this study, and deserves further investigation.*

line 40. The phrase, ". . .implications consist in a. . ." is awkwardly phrased. Perhaps revise to say rather: "Our results indicate a potentially relevant contribution. . ."

*Response: We have adopted the reviewer suggestion.*

line 58. Include "concentrations" after OH2

*Response: Changed.*

lines 83-84: Paragraph too long. I suggest inserting a paragraph break between these lines

*Response: Changed.*

lines 128-134: It appears that J-values for NO2 photolysis were calculated using clean flow tubes (i.e., in the absence of a coating). Therefore, one is assuming that the Jvalues for the clean tube are the same as those for a coated tube. Does the IC coating on the coated-wall flow tube attenuate the light enough to invalidate this assumption? If the IC coating does attenuate the light transmitted through the flowtube, how could this bias interpretation of the results?

*Response: This is not correct. The actinometry was measured in a clean glass tube and in a citric acid coated flow tube – the IC absorption in the actinic window (see Fig. S3) is very small. See also our response to Reviewer #1.*

line 253 (and other places where the term "H-donor" is used): The authors might want to clarify that the VOCs are H-atom donors, rather than proton donors as in the case of a Bronsted acid.

*Response: We have clarified this. The following sentence was added to Section 1, line 91. The sentences after were made a new paragraph.* "The citric acid and limonene are H-atom donors (referred to as H-donor from hereon), rather than proton donors as in the case of a Bronsted acid. In particular, the transfer of the H-atom leads to the formation of an alkyl-radical species. The H-atom transfer thus has the same effect as an H-atom abstraction reaction by Cl or OH radicals."

lines 314-318: It is not clear to me: Do the authors think that the NO to NO2 conversion by HO2 is occurring in the organic surface film or in the gas phase? How can they be sure?

*Response: As stated above (see replies to reviewers 1, and new Figure added to the SI), the HO$_2$ formation occurs at the surface and in the bulk of the condensed phase. It is somewhat improbable that NO reacts with HO$_2$ in the condensed (due to low NO solubility); formation of NO$_2$ may occur at the interface and in the gas phase. The comparison between the AFT and CWFT data suggests that the NO to NO$_2$ conversion is happening in the gas-phase, since this is a very fast reaction, apparently observed with higher efficient in the system with the lower total surface area (AFT).*

lines 319-323: The concentration of citric acid [CA] is in such excess relative to the imidazole. Could it not be assumed that [CA] is constant over the course of the experiments? In that case, why not just plot [HO2] vs [IC] instead of vs [IC]x[CA].

*Response: CA is constant throughout all experiments, and thus it could be presented either way. We prefer to show the product of IC x CA due to CA representing an H-atom donor, and thus plays a crucial role in the mechanism.*

lines 344-347: The authors discuss the role of coating viscosity on photochemistry. A recent article by Hinks et al. (PCCP, 2015, DOI: 10.1039/c5cp05226b) discusses this effect nicely and should be cited here.

*Response: Thanks for pointing this out. The reference has been added. "Hinks et al. (2016) observed that the photodegradation rate of their studied secondary organic material increases with increased RH. This suggests that the motion of the molecules in a viscous film at a low RH is hindered and thus results in a lower photochemical reaction."*

line 361: Cannot also H2O photophysically quench the triplet excited state? Is this important for this system?

*Response: In laser photolysis experiments (not reported here), it was shown that oxygen may quench the triplet state of IC (without suppressing it even in oxygenated solutions – see figure below) and that the decay rate was first order in the H-donor. This is strongly suggestive of a minor role (if any) of the products of that reaction on the lifetime of the triplet state of IC*

[Figure]

Figure 5-2 Triplet-triplet absorption of IC in water-acetonitrile (3:2);
(a) effect of degasing, (b) in the presence of cyclohexene.

[Figure]

*Figure 5-3 The change in the first order hate constant of IC for different limonene concentration; (a) transient absorption of excited triplet state of IC (0.25 m M). In the presence of (4 mM-Red), (8 mM – Blue) and (12 mM-pink) limonene; (b) first order hate constant as a function of limonene concentration.*

lines 403-406: The authors mention that the photosensitizer 4-BBA behaves similar to IC. I would like to see this data included in the SI.

*Response: The figures below have been added to the SI. The reader is reminded to see this Figure in the SI in Section 3.1.2.*

[Figure]

lines 426: With respect to ozone formation in the flowtube and aerosol systems, did the authors measure ozone and can they confirm that it was not observed in the system? Lifetimes of ozone were indicated for these experiments, do they account for heterogeneous loss as well?

*Response: Ozone formation was measured to be ca. 20 ppbv in the AFT experiments in presence of NO. It was produced due to the significant extent of NO₂ photolysis during these experiments (operated with long residence times (ca. 20-50 min) and led to the secondary chemistry mentioned above (and in the manuscript). Secondary chemistry and O₃ formation were suppressed in the CWFT experiments (1 ppm NO, limited NO₂ photolysis over 2 sec residence time). We estimate the upper limit for the O₃ concentration < 0.5 ppby O₃. Attempts to measure O₃ during selected experiments showed it below the detection limit. This is the primary reason why we believe the CWFT provides more quantitative determinations of P_{HO2}.*

Figure 8: Focusing on the "IM-C-OH(dot)" intermediate, should not the dot be centered on the carbon and not the alcohol H?

*Response: This has been changed. Write as IM-C(dot)-OH.*